# Pathophysiological Implications of Interstitial Cajal-like Cells (ICC-like) in Uterus: A Comparative Study with Gastrointestinal ICCs

Laura López-Pingarrón [1,*], Henrique Almeida [2,3,4,*], Desirée Pereboom-Maicas [1] and Joaquín J. García [1]

1   Department of Pharmacology, Physiology and Legal and Forensic Medicine, Faculty of Medicine, University of Zaragoza, 50009 Zaragoza, Spain; pereboom@unizar.es (D.P.-M.); jjgarcia@unizar.es (J.J.G.)
2   i3S—Instituto de Investigação e Inovação em Saúde, Porto University, 4200-135 Porto, Portugal
3   Department of Biomedicine, Faculty of Medicine, Porto University, 4200-319 Porto, Portugal
4   Department of Obstetrics and Gynecology, Hospital-CUF Porto, 4100-180 Porto, Portugal
*   Correspondence: lauralop@unizar.es (L.L.-P.); almeidah@med.up.pt (H.A.)

**Abstract:** The main function of interstitial cells of Cajal (ICCs) is to regulate gastrointestinal peristalsis by acting as a "pacemaker" cell by generating spontaneous slow electrical waves. In 2005, electron microscopy revealed a cell type similar to ICCs (ICC-like) outside the gastrointestinal tract, with contractile activity and c-Kit$^+$ immunohistochemistry shared with ICCs. Among the locations where ICC-like cells have been observed, it is in the uterus where they have a significant functional and pathophysiological role. These cells are involved in obstetric phenomena of contractile action, such as ascending sperm transport, embryo implantation, pregnancy, delivery, and the expulsion of menstrual debris. Within the pathophysiology related to these cells, we find obstetric alterations such as recurrent miscarriages, premature deliveries, abolition of uterine contractions, and failures of embryo implantation, in addition to other common conditions in the fertile age, such as endometriosis and leiomyoma.

**Keywords:** interstitial cells of Cajal; interstitial cells of Cajal-like cells; pacemaker; slow electrical waves; ultrastructure; immunohistochemistry; c-Kit; female genital tract; uterus; myometrial contractions; pregnancy; childbirth; abortion; preterm labor; endometriosis; leiomyoma

## 1. Introduction

### 1.1. Historical Background of ICC

In the 19th century, using light microscopy, the Spanish scientist Santiago Ramón y Cajal discovered a new cell lineage at the level of the nerve plexuses of the digestive system, located between neurons, glial cells, and smooth muscle cells [1]. Apart from this location, he also described them in Lierberkühn's glands, salivary glands, and the blood vessels of the pancreas [2,3]. Ramón y Cajal called them "interstitial neurons", due to their structural and staining similarity to neuronal cells [1,2], whose main function resided in the regulation of the motility of the digestive tract [4]. However, this discovery was not recognized by the scientific community until half a century later [5].

In the 1970s and 1980s, electron microscopic (EM) observations of these cells allowed a more precise definition of their ultrastructural characteristics [6]. With additional knowledge provided by immunohistochemistry, it was possible to distinguish them from other, morphologically similar cells, such as neurons, glial cells, smooth muscle cells, macrophages, and fibroblasts. Once this distinction was made, the "interstitial neurons" of Santiago Ramón y Cajal became known as interstitial cells of Cajal (ICCs) in honor of their discoverer [4].

### 1.2. ICC at the Gastrointestinal Tract Level

ICCs have been the subject of multiple studies. They are mesenchymal cells exhibiting typical ultrastructural, immunohistochemical, and functional characteristics, which are detailed below [5,6].

### 1.2.1. Ultrastructural and Immunohistochemical Characteristics

ICCs of the wall of the gastrointestinal tract form cell networks with connections to the enteric nerve plexuses. They are classified into three major subtypes: myenteric ICCs (ICC-MY), intramuscular ICCs (ICC-IM), and submucosal ICCs (ICC-SM). ICC-my are located at the level of the myenteric (Auerbach's) plexus, between the longitudinal and circular muscle layers; ICC-IM are located in the thickness of both muscular layers; and ICC-SM constitute the cellular network located in the submucosal (Meissner) plexus [5] (Figure 1).

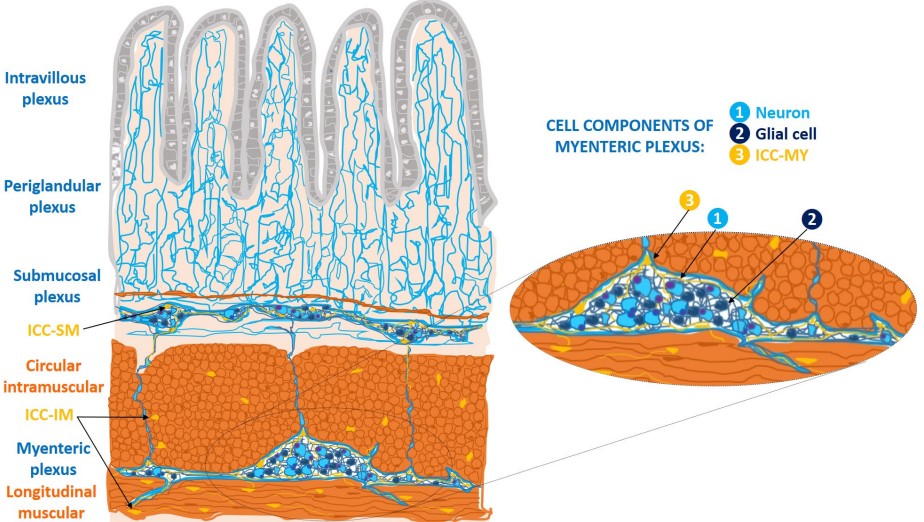

**Figure 1.** Location of cellular subtypes of interstitial cells of Cajal (ICCs) in the wall of the digestive tract (ICC-MY, ICC-IM, ICC-SM), based on a schematic drawing made by Santiago Ramón y Cajal [1]. ICC-MY: ICCs of the myenteric plexus, between the longitudinal muscle fibers and the circular muscle fibers; ICC-IM: ICC intramuscular at the level of both muscular layers; ICC-SM: ICCs of the submucosal plexus, and fibers of submucosal plexus, connecting with the myenteric plexus. The periglandular plexus in the Lieberkühn's glands; and the intravillous plexus in the intestinal villi. In detail: the components of myenteric plexus, in which there is a ganglion constituted by neurons, glial cells, and the ICC-MY [5].

EM visualization of ICC subtypes at the different locations shows that they share similar ultrastructural characteristics. They have spindle-shaped or oval cell bodies with long cytoplasmic extensions of 100 μm or more, which may branch into secondary and tertiary extensions [6]. Their plasma membranes have surface caveolae and a discontinuous basal lamina. The cytoplasmic organelles include abundant mitochondria, smooth endoplasmic reticulum cisternae, microtubules, fine and intermediate cytoskeletal filaments, free ribosomes, a discrete Golgi apparatus, rough endoplasmic reticulum and lysosomes. Also, these cells present gap-junction type membrane proteins that allow them to establish intercellular connections with neighboring smooth muscle cells [6].

Beyond their recognized ultrastructural features, ICCs present a series of defining staining and immunohistochemical characteristics [5]. They show positive staining for methylene blue and osmic acid/zinc iodine and, in immunohistochemical assessment, ICCs stain positively for the membrane receptor c-Kit [5]. This is a tyrosine kinase–type transmembrane receptor that binds to stem-cell growth factor (SCF), thus, regulating cell proliferation and division. It is encoded by the *KIT* proto-oncogene and plays a fundamental

role in the transduction of growth and repair signals [7]. Antibodies to c-Kit and its epitope CD117 are used to immunolabel ICCs, which are thus considered CD117/c-Kit$^+$ cells [5]. However, it must be taken into account that this marker is not exclusive to ICCs but shared by mast cells, mesenchymal stem cells, melanocytes, and germ cells, among others [8]. Recently, the chloride (Cl$^-$) channel activated by calcium (Ca$^{2+}$) Ano-1 was recognized as a new and highly specific immunohistochemical marker for ICCs of the gastrointestinal tract [7].

### 1.2.2. Functions

Current knowledge relates the three main functions of these cells to their gastrointestinal location: pacemaking for the peristalsis of the digestive tract, enteric neurotransmission and mechanoreception, and an important role in regulating the muscle tone of the wall [5]. First, ICCs' pacemaker properties refer to their involvement in the regulation of gastrointestinal peristaltic activity, as they have the capacity to generate spontaneous, slow electrical waves in the gastrointestinal tract [9,10]. Those slow electrical waves, whose fine mechanism is unknown, are associated with cytoplasmic oscillations of intracellular Ca$^{2+}$ ions, possibly by releasing Ca$^{2+}$ from intracellular stores. This electrical activity is transmitted to adjacent ICCs in the cell network, where it establishes the action potentials that trigger the gastrointestinal tract's regular contractility [11].

Second, ICCs participate in enteric neurotransmission by receiving excitatory and inhibitory neuronal stimuli from the enteric nervous system and transmitting them to neighboring smooth muscle cells. They thus stand in the middle of a functional neuromuscular network, establishing synapses with neurons and communicating with smooth muscle cells through gap junctions (Figure 2). By intermediating action potential transmission to adjacent muscle layers, ICCs trigger contractile activity in the gastrointestinal tract without the need for direct contact between enteric neurons and muscle fibers [5]. The neurotransmitter used by this neuromuscular network is nitric oxide (NO), a non-adrenergic, non-cholinergic inhibitory biomolecule synthesized from L-arginine by NO synthase (NOS). Released in response to nervous stimulation of the myenteric plexus, NO causes relaxation of the smooth muscle of the gastrointestinal wall. Thus, it regulates the peristaltic reflex as well as the muscle tone of the gastrointestinal sphincters, subordinate to ICCs [12].

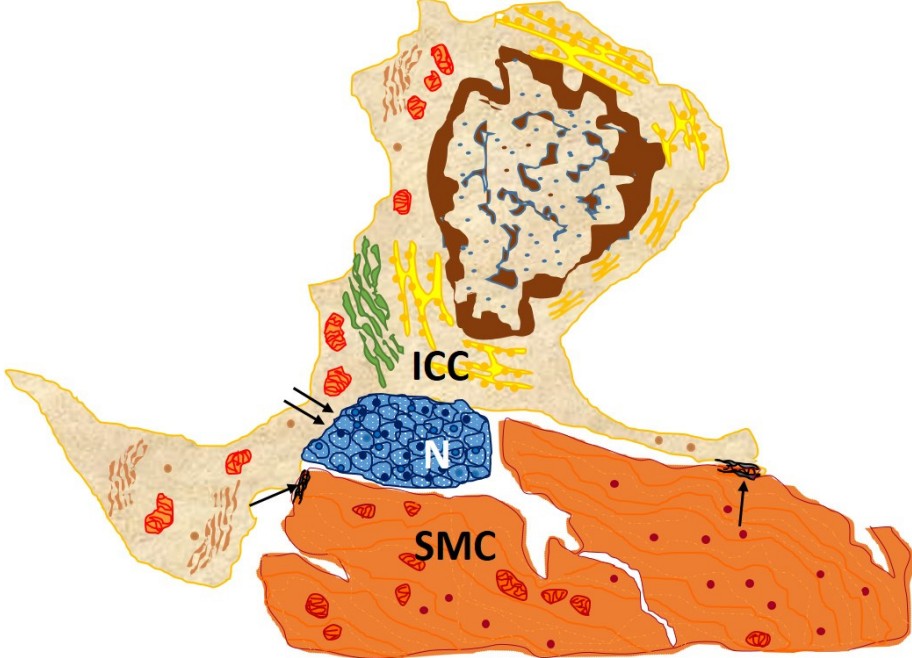

**Figure 2.** Drawn from EM image from the circular muscle layer of rabbit colon showing an interstitial cell of Cajal (ICC), with an elongated body and three processes located in the connective interstitium,

which contacts, through gap junction, between both its body and processes and two SMC (arrows) and nerve endings (double arrows). This cell owns the typical features of an intramuscular ICC: conspicuous Golgi apparatus and several rough endoplasmic reticulum cisternae; basal lamina is thin and discontinuous [13].

Finally, ICCs also function as mechanoreceptors. ICCs collect continuous mechanical signals, originating in the distensibility of the gastrointestinal walls, which occurs during food propulsion, digestion, and waste elimination. In response to this regular chronotropism of the stretching of the gastrointestinal tract wall musculature, ICCs adjust gastrointestinal activity by regulating the frequency of the contraction and the triggering of slow electrical waves [12].

### 1.2.3. Disfunctions

Taking into account these three primary functions described above, it is understandable that changes in ICC numbers are associated with gastrointestinal motility disorders [14]. Several pathologies of childhood and old age have been associated with variations in the number of ICCs, as demonstrated by c-Kit immunolabeling. This is the case in pediatric conditions such as Hirschprung's disease, the most prevalent congenital gastrointestinal motility disorder, a functional intestinal obstruction resulting from a defect in the intrinsic innervation of the intestine. Since ICCs have a central role in gastrointestinal peristalsis and enteric neurotransmission, their quantitative decrease may explain the motility dysfunction that occurs in this pediatric disease [15]. At older ages, a decrease in ICC numbers has been related to disorders such as acquired hypoganglionosis, diabetic or idiopathic gastroparesis, intestinal pseudo-obstruction, and chronic constipation [14,16]. Finally, and independently of age, megacolon associated with Chagas disease could also be explained by a lower density of ICCs. This is a "megasyndrome" defined as a chronic dilatation of the colonic segment, associated with infection by the *Trypanosoma cruzi* parasite and the destruction of the enteric nervous system [17].

The pathophysiological correlation of ICCs with disease has been studied in gastrointestinal stromal tumors (GISTs). In GISTs, mutation of the *c-kit* gene leads to SCF-independent activity, triggering cell division and genomic instability that result in neoplasia. ICC-derived GISTs can be identified by their immunohistochemical positivity for vimentin, CD34, and c-Kit [18,19]. These markers have allowed us to understand the pathogenesis of GIST and have made it possible to develop an effective treatment targeting c-Kit.

There are other functions linked to ICCs that have given rise to controversy and are still a matter of debate. This is the case for the plastic property by which ICCs have the capacity to adapt to adverse environmental conditions [20]. They appear to survive different noxae by changing their phenotype to that of smooth muscle cells and/or intermediate cells, or fibroblasts, thus losing the capacity to generate slow waves. When the threat is overcome, they return to their previous, normal morphology and their intrinsic electrical activity is restored [20].

## 2. ICC-like Cells Outside the Digestive System

Employing the EM in studies outside the gastrointestinal tract, Popescu and colleagues described a cell type similar to ICCs, which they named Interstitial Cajal-like cells (ICC-like) [21]. This appellation was selected because the new cells share some ultrastructural, immunohistochemical, and functional characteristics with ICCs. Later, several researchers modified the designation of ICC-like cells to telocytes (TCs), to emphasize the qualitative and conceptual distinction. ICC-like cells have been observed in numerous extraintestinal locations over the last 15 years, the most recognized being the male and female reproductive systems, placenta, mammary glands [22], cardiovascular system, urinary bladder, prostate [23], liver [24], lungs [25], and skin [26].

## 2.1. Ultraestructural Characteristics

Broadly speaking, ICC-like cells share with ICCs some cytoplasmic features and membrane properties. The typical cytoplasmic extensions define the stellate morphology of ICC-like cells. Here there are highlighted some of their morphologic characteristics:

- They are of great length, tens to hundreds of $\mu$m, and considered one of the longest structures in the human body [27];
- They vary in number from one to five, there frequently being two or three per cell. Thus, the three-dimensional appearance of ICC-like cells is that of a polyhedron with a variable number of vertices according to the number of extensions;
- They are thin, <0.2 $\mu$m, and moniliform in appearance, with dilations along their length harboring $Ca^{2+}$ channels and cytoplasmic organelles [27].

These specific characteristics of the ICC-like cells' extensions distinguish them from other cytoplasmic cellular extensions such as dendrites and axons, precluding any type of relationship between ICC-like cells and neurons. In turn, they constitute a structural feature that differentiates ICCs and ICC-like cells from other similar cells, such as fibroblasts, myofibroblasts, and dendritic cells [13].

## 2.2. Immunohistochemical Characteristics

Apart from CD117/c-Kit positivity shared with ICCs of the gastrointestinal tract, ICC-like cells present in their membrane a series of proteins (CD34, vimentin, platelet-derived growth factor receptors alpha and beta [PDGFR-$\alpha$ and $\beta$], caveolin-1, CD44, stem cell antigen-1 [Sca-1], Nanog and octamer-binding protein 4 [Oct-4]) that make their immunohistochemical recognition distinctive. Indeed, the most commonly used markers for the study of ICC-like cells are CD34 and c-Kit, due to a number of ICC-like cells having been recognized by the double stain for both antibodies, showing an irregular pattern on the cell body [21]. Therefore, the morphological features previously described together with immunolabeling allows us to confirm or rule out ICC-like cell identity. It is a pending task to discover a distinctive marker for ICC-like cells, beyond c-Kit, to facilitate further research and scientific progress [28].

## 2.3. Functions of ICC-like Cells

The main recognized function of ICC-like cells is intercellular signaling, taking advantage of their strategic position between blood vessels, nerves, and other adjacent cells. This paracrine/juxtacrine type of intercellular signaling, mediated by the shedding of microvesicles in the form of exosomes, ectosomes, and multivesicles, occurs in a homocellular and heterocellular manner [28,29]. In the case of homocellular communication, it takes place between two extensions of ICC-like cells or between one extension and the body of an ICC-like cell. In heterocellular signaling, ICC-like cells communicate with different cell types, such as fibroblasts, myofibroblasts, pericytes, endothelial cells, neurons, stem cells, macrophages, mast cells, eosinophils, lymphocytes, plasma cells, Schwann cells, cardiomyocytes, and smooth muscle cells [22].

These heterocellular communications allow ICC-like cells to modulate the immune response, regulate blood flow, and contribute to tissue regeneration and repair, the organization of the extracellular matrix, cell migration, and the maintenance of tissue homeostasis. In addition, by producing vascular endothelial growth factor (VEGF), they play an angiogenic role; moreover, by producing abundant superoxide dismutase (SOD2), they have antioxidant properties. Considering all these functions, the use of ICC-like cells in regenerative medicine as a therapy is a possibility [22].

ICC-like cells share with ICCs electrical capacity and can influence contractile activity, despite some controversy on the intrinsic function of ICC-like cells in the generation of slow electrical waves and their pacemaker function [27].

The capacity of ICC-like cells to generate electrical activity associated with increases in intracellular $Ca^{2+}$ concentration has been questioned, because isolated ICC-like cells do not have spontaneous intracellular $Ca^{2+}$ oscillations or electrical activity. However, it is

likely that the ICC-like cellular network arrangement provides amplification sufficient to generate detectable action potentials [30]. Oscillations in intracellular $Ca^{2+}$ concentrations have been demonstrated in myometrial tissue as the starting point for the propagation of contractile signals in the network of ICC-like cells [31]. Therefore, several views support the pacemaker role of ICC-like cells in the generation of their own electrical activity [30]. It is a variable cellular capacity dependent on the contractile activity linked to each location, and mainly present in the urinary tract and male genital tract [32].

## 3. ICC-like Cells in Female Genital Tract

In the female genital tract, ICC-like cells have been linked to the generation of slow electrical waves at the tubal level that trigger the contractility necessary for egg transport [33]. Similarly, in the myometrium, electrical waves linked to ICC-like cells have possibly been detected [30]. However, in this topic there is still a great deal of uncertainty to be resolved. Some hypotheses advocate that it is, above all, in organs with regular contractile activity, such as the pancreas and other exocrine secreting glands, that this pacemaker function of ICC-like cells is active [2,3,29].

The presence of ICC-like cells in the myometrium and mammary glands was described by Popescu et al. [21]. Currently, in the female genital tract, ICC-like cells are also documented in the vagina, fallopian tubes, ovaries, and placenta. In each of these locations, the density and functions of ICC-like cells may vary according to local physiological needs [21]. Most of the studies that relate ICC-like cells to the female genital tract focus on the uterus, where quantitative changes in those cells are associated with functional and pathological contractility phenomena [28].

### 3.1. Ultraestructural and Immunohistochemical Features

In the uterus, ICC-like cells are located in the endometrium and myometrium [21]. The myometrium is composed of three poorly differentiated layers of smooth muscle bundles: the inner longitudinal layer, the vascularized middle circular layer, and the outer longitudinal layer [34]. It is between the muscle bundles of the different layers that the myometrial ICC-like cells are found [21]. The endometrium is the innermost layer of the uterus and the one with the greatest number of histological components. It is made up mainly of connective tissue, glands, and stromal cells that, during gestation, are transformed into decidual cells. During fertile life, the endometrium differentiates into the basal or deep layer, which is highly vascularized and regenerating, and the functional or superficial layer, which is shed with the menstrual period [34]. It is between the stromal glands of both endometrial layers that endometrial ICC-like cells are identified [35].

As in other locations, uterine ICC-like cells express plasma membrane CD117/c-Kit protein, which is frequently targeted for immunohistochemical detection (Figure 3).

Other frequent targets for immunolabeling in the uterus are CD34 and PDGFR-$\alpha$ and $\beta$ proteins. CD34 is preferentially located in the plasma membrane of uterine ICC-like cells' cytoplasmic extensions, whereas PDGFR-$\alpha$ and $\beta$ are more prevalent in the cell body. The complete immunohistochemical pattern presented by uterine ICC-like cells is as follows: $\alpha$-SMA$^+$, CD44$^+$, vimentin$^+$, Sca-1$^+$, CD117/c-Kit$^+$, and connexin 43$^+$, in addition to CD34$^+$ and PDGFR-$\alpha$/$\beta^+$ [21].

Uterine ICC-like cells exhibit plasma membrane estrogen receptor alpha (ER$\alpha$) and progesterone receptor-A (PR-A), which, interestingly, are also found in the uterine tubes [21]. This common phenotype suggests that uterine ICC-like cells' activities are modulated by levels of circulating sex hormones, which further suggests an additional cyclic regulatory role in organ response. It also makes these cells a likely target for new hormone therapies [28].

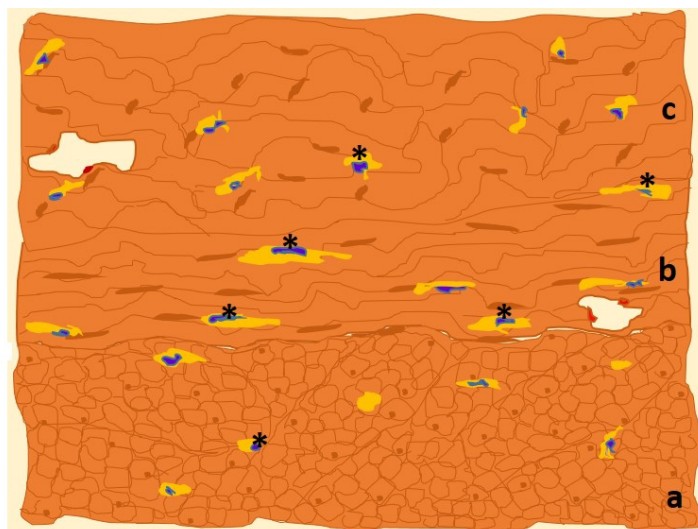

**Figure 3.** Drawn of MO image of a human uterine myometrial tissue sample in which immuno-histochemistry for c-Kit has been used. The immunolabeled ICCs-like cells/telocytes (*, in yellow) are identified and distributed along the three constituent layers of uterine smooth muscle: outer longitudinal (a), middle circular (b) and inner longitudinal (c) layers [36].

Finally, uterine ICC-like cells present two main types of transmembrane channels: T-type $Ca^{2+}$ channels (Cav3.1 and Cav3.2) and small-conductance $Ca^{2+}$-activated potassium ($K^+$) channels (SK3), which are both involved in uterine contractility [21,37]. The presence of $Ca^{2+}$ channels in ICC-like cells' plasma membranes and their involvement in the contractile activity of the uterus have been demonstrated by administering mibefradil®. This is a T-type $Ca^{2+}$ channel antagonist drug which, by inhibiting the contractility linked to these receptors, is capable of suppressing bioelectrical signals and uterine contractile forces [38].

Table 1 summarizes the main morphological, ultrastructural, and immunohistochemical characteristics of enteric ICCs and myometrial ICC-like cells, mainly described versus the endometrial cells up to this point, and incorporates data on their intercellular relationships.

**Table 1.** Comparison of morphological, ultrastructural, and immunohistochemical features of GI tract-like ICCs with myometrial uterine ICCs-like cells, in addition to the intercellular communications associated with each (modified from [30]).

| Cytological Issues | Enteric ICC | Myometrial ICC-like Cells |
|---|---|---|
| Shape: | | |
|    cell | Oval or spindle | Spindle or stellate |
|    nucleus | Oval, mostly euchromatic | Oval, heterochromatic |
| Mitochondria | +++ | ++ |
| ER [1]: | | |
|    smooth | ++ | + |
|    rough | + | + |
| Golgi complex | + | + |
| Filaments: | | |
|    thin | + | + |
|    intermediate | ++ | + |
|    Microtubules | + | + |
| Calcium releasing units | | + |
| Caveolae | + | + |
| Basal lamina | | +- |
| Gap junctions | + | + |

**Table 1.** *Cont.*

| Cytological Issues | Enteric ICC | Myometrial ICC-like Cells |
|---|:---:|:---:|
| Intercellular contacts: | | |
| nerve endings | ++ | + |
| blood vessels | | + |
| immune cells | | +++ |
| muscular cells | + | + |
| other interstitial cells | + | + |
| Immuno-histochemical markers | c-Kit, Ano-1 | c-Kit, CD34, PDGFR-$\alpha$, smooth muscle actin ($\alpha$-SMA), CD44, vimentin, Sca-1, connexin 43 |

[1] ER: endoplasmic reticulum. +: low presence; ++: medium presence; +++: high presence.

### 3.2. Functions of Uterine ICC-like Cells

The fundamental role of uterine ICC-like cells is the same as that of ICC-like cells in other extragenital locations—to trigger contractile activity [30].

In utero, variations in myometrial contractility are a part of both normal pregnancy and delivery and their pathological forms of miscarriage and premature delivery. These events occurring in the female genital tract, together with other pathological findings such as leiomyomas and endometriosis, have a certain relationship with uterine ICC-like cells, as detailed below [21,37].

### 3.2.1. ICC-like Cells Implications in the Physiology of Pregnancy and Labor

The myometrial contractile activity associated with uterine ICC-like cell functioning favors the upward transport of sperm prior to fertilization, embryo implantation, variations in uterine capacitance during pregnancy or delivery, and the expulsion of menstrual debris [37]. This contractility triggered by uterine ICC-like cells is associated not only with intracellular oscillations in $Ca^{2+}$ concentrations, but also with their quantitative variations, which are modulated by signaling through their membrane proteins [21]. These include the immunohistochemical markers connexin 43 and CD117/c-Kit, the hormone receptors ER$\alpha$ and PR-A, and the transmembrane channels Cav3.1, Cav3.2, and SK3 [28–30].

During the course of pregnancy, estrogen and progesterone expression varies, with maximal estrogen levels and minimum progesterone concentrations occurring at the time of labor; in fact, it is the increase in estrogens, oxytocin, and prostaglandins that triggers labor contractions [39]. Moreover, to promote them, enhanced expression of myometrial oxytocin receptors [40], and doubling of circulating uterine prostaglandin should occur [39]. Effective contractions become increasingly painful and recurrent, in contrast to other, "false" or ineffective sporadic (Braxton-Hicks) uterine contractions that do not lead to labor [41].

In the physiology of gestation and childbirth, some of the hormonal alterations mentioned are related to the activity of uterine ICC-like cells. In fact, the plasma-membrane protein components of uterine ICC-like cells, such as ER$\alpha$ and PR-A receptors and the immunomarker connexin 43, not only increase in expression as gestation progresses, they also play a role in the physiology of uterine contractions of labor [28,30].

Labor is a complex physiological process, defined by numerous hormonal interactions that trigger uterine contractions that, ultimately, lead to fetal expulsion. At "term"—considered as the time interval between 37 and 40 weeks plus 6 days of gestation—there is an imbalance between the myorelaxant hormones that favor pregnancy (progesterone, nitric oxide, catecholamines, and relaxin) and those that trigger labor (estrogens, oxytocin, and prostaglandins), in favor of the latter.

ER$\alpha$ and PR-A promote overexpression of T-type $Ca^{2+}$ channels in the ICC-like cell membrane, triggering the continued uterine contractions characteristic of term [27]. The localization of the gap junction protein connexin 43 in uterine ICC-like cells allows the establishment of multiple connections with surrounding myocytes. Together with the

involvement of estrogens, oxytocin, and prostaglandins, the result is the development of regular forceful contractions.

It is argued that the peculiar widespread localization and gap-junction nature of connexin 43 at the time of parturition provides the means for triggering a synchronous contraction of all uterine smooth muscle cells, constituting a "functional syncytium" [30]. However, this view has been challenged and an alternative model of "biphasic" uterine contraction, defined by a fast electrical wave followed by a slower one, has been advocated. In this model, there is rapid electrical conduction through most of the myometrium, followed by a slower phase of recruitment of the remaining number of myocytes via a "calcium wave" (Young's biphasic theory); myometrial ICC-like cells have been identified as responsible for the second, slow phase of contraction [42]. Both models fail to provide a complete physiological explanation of the contractile moment of labor, the mechanism that generates the action potential in the myometrium, the reason for the selective recruitment of smooth muscle cells in the fast phase, or why Braxton-Hicks contractions are ineffective [30].

Once delivery and the muscular stress it entails are over, the uterus returns to its usual state and does so by means of vigorous contractions, which reduce uterine capacitance and increase the number or density of myometrial ICC-like cells. This change has likely evolved to recover pre-pregnancy myometrial tone and dimensions [43].

In pregnancy, myometrial ICC-like cells are less densely distributed than they are in pre-pregnancy and even sparser than in the post-partum period. In fact, the non-pregnant uterus has less contractile activity than the postpartum uterus, but more than the pregnant uterus; this change, possibly related to ICC-like cell density, prevents unwanted preterm uterine contractions and acts as a protector of the fetus until term [27,43].

Beyond the differential density of myometrial and endometrial ICC-like cells, their expression of the transmembrane Cav3.1, Cav3.2, and SK3 channels influences contractility in the two uterine states (Table 2). The expression of Cav3.1 channels is equal in the pregnant and non-pregnant uterus, while Cav3.2 expression is lower in non-pregnant myometrium. In addition, their locations in the cell differ: Cav3.1 channels are found mainly on the surface of the extensions, whereas Cav3.2 channels localize to the cell body's plasma membrane [44].

**Table 2.** Differences between non-pregnant uterus, pregnant uterus, and postpartum uterus according to quantitative variations of the number of endometrial and myometrial ICCs-like [43], and expression in the membrane of Cav3.1, Cav3.2 [44], and SK3 channels [45].

|  | Non-Pregnant Uterus | Pregnant Uterus | Postpartum Uterus |
|---|---|---|---|
| Endometrial ICC-like cells | + | ++ | + |
| Myometrial ICC-like cells | ++ | + | +++ |
| Cav3.1 | ++ | ++ |  |
| Cav3.2 | + | ++ |  |
| SK3 | ++ | + |  |

+: low presence; ++: medium presence; +++: high presence.

Moreover, SK3 channels exhibit greater expression in the plasma membranes of ICC-like cells and in the endothelium from the non-pregnant condition. In pregnancy, they maintain both localizations but at a lower expression intensity in the uterine ICC-like cells' plasma membranes. These patterns of expression of $Ca^{2+}$-dependent transmembrane channels in the pregnant uterus favor the mechanical stretching of the uterus during pregnancy and a reduction in myometrial uterine contractility [45].

Another mechanism implicated in myometrial contractility involves the cell membrane marker CD117/c-Kit. Apart from being used to identify ICCs, it is targeted by its antagonist imatinib® [46] to reduce the frequency and amplitude of myometrial contractions associated with these cells. The compound does not block them entirely because ICC-like cells can maintain uterine contractility independently of c-Kit, under hormonal influence [27].

Another remarkable physiological feature of uterine ICC-like cells, independent of myometrial contractility and common to other locations, is their endocytic property. They have the capacity to not only transmit intercellular signals by generating ectosomes, exosomes, and multivesicles, but also to receive them through endosomes, thus establishing bidirectional communication with neighboring cells [47]. Among the cells with which they exchange signals, there is a predominance of immune system constituents, especially macrophages, thus explaining the fundamental role of uterine ICC-like cells in immune surveillance [21].

There is, thus, a considerable amount of information favoring the existence of uterine ICC-like cells, with distinct specific localization and distribution patterns of molecules. These favor their involvement in uterine contractions and in local signaling that regulates the frequency of contractions and the surveillance of signals related to labor conditions.

While myometrial ICC-like cells have a strong involvement in the uterine contractions, endometrial ICC-like cell density is more related to intercellular signaling. In contrast to myometrial ICC-like cells, endometrial ICC-like cells are found in lower numbers in the non-pregnant and the post-partum uterus than in the pregnant uterus [27]. A possible explanation is that non-pregnant endometrial stroma is so compact as not to require many intercellular communications through cell extensions, whereas during pregnancy, uterine enlargement fosters the need for such communications and the expansion of ICC-like cell connections [28,48].

### 3.2.2. Physiopathological Implications

Due to their distribution, uterine ICC-like cells are implicated in gynecological and obstetric disorders. They are linked to recurrent miscarriages, premature deliveries, abolition of uterine contractions, and implantation failures. Beyond obstetric pathophysiology, associations of these cells with pathological conditions of the non-pregnant uterus such as endometriosis and leiomyoma have also been described [21,37].

- Gynecological disorders

Uterine fibroma (or leiomyoma), is the most common benign tumor of the female genital tract: worldwide, around 50% of women present it during their fertile life [8]. Clinically, these tumors manifest as chronic pelvic pain, abundant bleeding, spontaneous abortions, and infertility, all associated with deterioration in the quality of life [8]. There is an imbalance in the three-dimensional organization of the uterine tissue in favor of an excessive increase in the number of fibroblasts and myocytes [37]. In addition, active angiogenesis favors nutrient supply and tumor growth, although microvascular density in leiomyoma is still lower than it is in healthy myometrium, and there are even small avascular fibromas [37], which creates an ischemic milieu that influences leiomyoma dimension. The new vessels formation and a marked local autonomic nervous component result in "pseudocapsules" [49].

Leiomyomas are described as steroid-hormone sensitive and even as hormone-dependent [36]. Estrogen receptors and, to a lesser extent, progesterone receptors, are overexpressed in myocytes [50], which under estrogen signaling release growth factors such as VEGF or PDGF. The enhanced nervous density in leiomyomas, greater than in healthy myometrium, has been demonstrated by means of positive immunoreactivity for markers of autonomic innervation such as protein gene product 9.5 (PGP 9.5) and inducible nitric oxide synthase (iNOS) [51].

Leiomyomas are tumors that contain uterine ICC-like cells, but smooth muscle cells and fibroblasts predominate [37]. The balance between these cell types oscillates according to variation in the dimensions of the tumor, which reflect the cell density. When smooth muscle cells and fibroblasts are disproportionately increased in number, this physiological balance is broken and uterine c-Kit$^+$ ICC-like cells decrease in number or disappear completely [52]. To explain their reduction, a theory of the "rupture of the physiological cellular equilibrium" was proposed [37]. As cells are sensitive to deficits in tissue perfusion, their density decreases. This in turn leads to decreased VEGF production, reduced

capacity for neovessel formation, and a shift from aerobic to anaerobic metabolism in myocytes [21]. These modifications result in the acquisition of an avascular phenotype similar to leiomyoma, which produces a more perfusional deficit [8].

A decrease in the number of uterine ICC-like cells also alters their involvement in local regulatory function, which is then relegated to myocytes, the most prevalent cell in the leiomyomatous condition [37], which now proliferate in an abnormally oriented and dysfunctional fashion [8].

Beyond the numeric decrement, ultrastructural changes in uterine ICC-like cells have also been reported and include mitochondrial enlargement, cytoplasmic vacuolization, and the formation of lipofuscin bodies.

ICC-like cells are known to have an intermediary role in tumor innervation, through signaling by peritoneal macrophages to synthesize iNOS, which is used as an immunohistochemical marker for autonomic innervation detection [53]. The close spatial relationship between uterine ICC-like cells and myometrial autonomic nerve fibers has been demonstrated by double-positive immunostaining for both iNOS and PGP 9.5 as well as CD34 in the same sample [51].

Interestingly, increased iNOS expression in leiomyoma reduces the frequency of tubal ciliary beating and the smooth muscle contractility of the oviducts, thus contributing to tubal disorders that can accompany infertility [54].

Finally, uterine ICC-like cells, by sharing intercellular signals with myometrial mast cells and myocytes, can participate in tissue remodeling and growth. This is because myometrial mast cells have surface receptors for SCF, a smooth-muscle-cell factor and modulator of mast cells' role in tissue remodeling. This contributes to more extensive consequences involving local inflammation and further leiomyoma growth [37].

Endometriosis is characterized by the presence of endometrial tissue outside the uterine cavity, for whose origin three theories have been proposed: retrograde menstruation, metaplasia of the germinal tissue, and metastatic dissemination [55,56]. To them, we can add a role for the local adrenergic nervous system in triggering chronic pelvic pain as a result of the enhanced estrogen content associated with the disorder [50,57].

In addition, an alteration in the intercellular signaling between uterine ICC-like and the cells of the immune system is also a likely component of endometriosis pathophysiology. ICC-like cells establish regular communication with the uterus, with either the neuromuscular network or cells of the immune system. This results, for example, in disturbances of IL-6, IL-10, IL1R1, and tumor necrosis factor alpha (TNFα) synthesis and secretion by peritoneal macrophages. The interaction may also involve tubal ICC-like cells [27,53].

- Obstetric disorders

Pathological conditions linked to pregnancy and childbirth are influenced by membrane components of the uterine ICC-like cells, together with the intercellular communications that they establish with adjacent cells. That is the case for recurrent abortions, abolition of uterine contractions, premature delivery, and implantation failure [37].

Recurrent or repeated spontaneous abortions are defined as a succession of three or more consecutive abortions of an embryo or fetus weighing 500 g or less [39]. These recurrent miscarriages may be due to multiple causes, including connexin-43 protein deficiency. This protein, apart from participating in the contractile activity linked to parturition, also favors the maturation of the decidua [28,29], whose cells differentiate from endometrial cells and are eliminated with menstruation. As the decidual tissue allows the exchange of nutrients, gases, and waste products with the fetus, lack of maturation of the decidua is related to nonviable gestation and recurrent miscarriages [34]. Connexin-43 malfunction also contributes to the abolition of uterine contractions during labor [58], so that additional actions for appropriate delivery are required [34].

Preterm deliveries are defined as those occurring before 37 weeks of gestation [39]. In these cases, it is desirable that myometrial ICC-like cells density decrease to prevent contractions and prematurity [28].

Embryo implantation failure results from defective intercellular signaling between ICC-like cells and cells of the immune system that is likely to destabilize immunosurveillance of the uterine environment. Miscarriage and endometriosis appear to result from the same defect. These three conditions also appear to derive from peritoneal macrophages' abnormal production of IL-6, IL-10, IL-1R1, and TNFα [53].

Other gynecologic or obstetric conditions such as acute salpingitis, breast cancer, tubal infertility, ectopic pregnancy, and preeclampsia may also involve changes in the modulatory effects of local ICC-like cells consequent to yet unclear mechanisms [29].

## 4. Conclusions

ICCs are c-Kit$^+$ cells located largely in the wall of the gastrointestinal tract where, through an extensive network, they generate slow electrical waves and exert a pacemaker function, regulating gastrointestinal peristalsis. Among the pathologies with which they have been associated, GIST tumors stand out.

On account of their characteristic ultrastructure, with long moniliform cytoplasmic extensions, and a predominance of positive CD34 and c-Kit immunohistochemistry, ICC-like cells are recognized in the uterine tissue, where they mediate immunosurveillance, myometrial regeneration, and contractility.

Uterine ICC-like cells' contractility is relevant to the physiology of pregnancy and childbirth. Quantitative oscillations of cell density and the ICC-like cell membrane receptors ERα, PR-A, and others, such as connexin 43, further contribute to the contractility of parturition. Finally, uterine ICC-like cells' intervention or density may contribute to the pathophysiology of leiomyomas, recurrent miscarriages, and preterm deliveries.

## 5. Perspectives

Knowledge about the functions of uterine ICC-like cells is growing, but is fragmentary in some aspects. Continued research is needed, especially in the following areas:

1. The lack of a specific immunomarker for ICC-like cells prevents their indisputable identification. Progress on this point is necessary to better understand the extent of ICC-like cell presence in uterine structures in particular, and in other anatomical sites;

2. Such mapping of ICC-like cells in healthy conditions will be of the utmost importance in verifying these cells' involvement and potential changes in a variety of disorders;

3. The identification and mapping of these cells will also provide important clues regarding the roles of genes encoding such markers and will guide strategies designed to unveil their functional properties;

4. Further in-depth knowledge from such studies will provide clues for the design of modulatory molecules with potential use in hormonal or tissue regenerative therapies. Information regarding specific applications is emerging, as exemplified by the potential use of the c-Kit tyrosine-kinase inhibitor imatinib in leiomyosarcomas of the uterus [59] and as an inhibitor of rhythmic human uterine contractions [60].

**Author Contributions:** Conceptualization, L.L.-P., H.A. and J.J.G.; methodology, L.L.-P., H.A., J.J.G. and D.P.-M.; writing—original draft preparation, L.L.-P., H.A. and J.J.G.; writing—review and editing, L.L.-P., H.A. and J.J.G.; supervision, H.A. and J.J.G. All authors have read and agreed to the published version of the manuscript.

**Funding:** This work was supported by a grant from the Gobierno de Aragón (grant No. B56_23D).

**Institutional Review Board Statement:** Not applicable.

**Informed Consent Statement:** Not applicable.

**Data Availability Statement:** Not applicable.

**Conflicts of Interest:** The authors declare no conflict of interest.

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
