# Peer review of "Pathophysiological Implications of Interstitial Cajal-like Cells (ICC-like) in Uterus: A Comparative Study with Gastrointestinal ICCs"

_cimb, doi:10.3390/cimb45090476_

Round 1

Reviewer 1 Report

In this review article, the authors discussed the interstitial cells of Cajal (ICCs), located largely in the wall of the gastrointestinal tract where through an extensive network. ICCS generate slow electrical waves and exert a pacemaker function, regulating gastrointestinal peristalsis. Among the pathologies with which ICCs have been associated, GIST tumors stand out. On account of their characteristic ultrastructure, with long moniliform cytoplasmic extensions, and a predominance of positive CD34 and c-Kit immunohistochemistry, ICC-like cells are recognized in the uterine tissue where they mediate immunosurveillance, myometrial regeneration, and contractility. Uterine ICC-like cells’ contractility is relevant to the physiology of pregnancy and childbirth. Quantitative oscillations of cell density and the ICC-like cell membrane receptors ERα, PR-A, and others such as connexin 43, further contribute to the contractility of parturition. Finally, uterine ICC-like cells’ intervention or density may contribute to the pathophysiology of leiomyomas, recurrent miscarriages, and preterm deliveries.

The review article was well-written and large amount of the relevant references have been cited. The figures are beautiful. This is a really good review.

Author Response

Thank you very much for your kind comments. We are very happy if this review have been interesting for you.

Reviewer 2 Report

The manuscript entitled as “Pathophysiological Implications of Interstitial Cajal-like cells (ICC-like) in Uterus” is a review that addresses Cajal-like cells in uterus. The manuscript is well-written as brings schemes and tables to illustrate and facilitate the reading. Surprisingly, there are feel reviews published approaching this topic. Below, the authors can find some comments and suggestions.

Major concerns:

In the reviewer’s opinion, the only major consideration is regarding the title itself. Almost half of the manuscript is about historic facts, gastrointestinal ICCs, and general ICCs outside the female genital tract, which are relevant information and are important to contextualize the topic. With that said, the reviewer suggests that the title should contain that it is a review about ICCs, with a focus on the uterus.

Minor concerns/corrections:

Line 53 – a ) is missing

Figure 1 caption: Location in instead of localization?

As a suggestion, create a subdivision (1.2.3.) to cite and describe the pathologies from intestinal system. Currently, this part of the text is in the function of the cells.

Add reasons to use cd34 as marker, line 186.

3.2.1. Start with pregnancy then, talk about labor?

3.2.1. The name of this division is too generic. The reviewer suggests an alteration for something related to the text. “ICC-like implications in the physiology of pregnancy and labor” for instance.

Minor editing of English language required.

Author Response

Major concerns:

In the reviewer’s opinion, the only major consideration is regarding the title itself. Almost half of the manuscript is about historic facts, gastrointestinal ICCs, and general ICCs outside the femalegenital tract, which are relevant information and are important to contextualize the topic. With that said, the reviewer suggests that the title should contain that it is a review about ICCs, with a focus on the uterus.

Response 1: Thank you very much for such interesting appreciation. We are totally agree and we have added another sentence to the tittle to better clarify what the review is about. The rewritten tittle is as follows: “Pathophysiological implications of Interstitial Cajal-like cells (ICC-like) in Uterus: a comparative study with gastrointestinal ICCs”.

Minor concerns/corrections:

Point 1:Line 53 – a ) is missing

Response 1: Thank you-we have added it after (ICC-IM) in line 54.

Point 2: Figure 1 caption: Location in instead of localization?

Response 2: We apologize for this mistake-we have corrected it in line 60.

Point 3: As a suggestion, create a subdivision (1.2.3.) to cite and describe the pathologies from intestinal system. Currently, this part of the text is in the function of the cells.

Response 3: Thank you so much for this comment. We have added a new subsection as you mentioned, which is called “1.2.3. Dysfunctions” in line 130.

Point 4: Add reasons to use cd34 as marker, line 186.

Response 4: In response to your request, we have highlighted the main reason for using CD 34, as supported by reference 21, because double staining for c-kit and CD34 demonstrated that both antibodies recognized several ICC-like cells, showing an irregular pattern in the cell body.

Point 5: 3.2.1. Start with pregnancy then, talk about labor?

Response 5: It is a very good point of view, so we have changed the paragraph about labor from line 308 to line 325, after the text “they also play a role in the physiology of uterine contractions of labor”.

Point 6: 3.2.1. The name of this division is too generic. The reviewer suggests an alteration for something related to the text. “ICC-like implications in the physiology of pregnancy and labor” for instance.

Response 6: We totally agree with you. This tittle suggested has been written for this division in line 303.

Comments on theQuality of EnglishLanguage: Minor editing of English language required.

The manuscript was edited by a Company for general proofreading and editing (San Francisco Edit, invoice No: 230198). We have highlighted the minor corrections in the new attached version.

Thank you very much for your advice and guidance. Your comments have really improve this manuscript.

Reviewer 3 Report

The interstitial cells of Kajal, from Santiago Ramón y Cajal who firstly discovered this new cell lineage at the level of the nerve plexuses of the gastrointestinal tract, Their main function is to regulate gastrointestinal peristalsis by acting as a "pacemaker" cell by generating spontaneous slow electrical waves.

In this manuscript, the authors have expertly described the ultrastructural, histological, metabolical and physiological aspects of ICC-like cells, and the mechanisms that regulate their activation, starting from gastrointestinal tract level to other organs, principally uterus, where they share the most important function of regulating cell/organ contractile action.

It is a very interesting, well written and logically structured review that describes the most recent knowledge about these cells, opening up new perspectives in the study/treatment of pathologies, such as recurrent miscarriages, and premature deliveries, as well as endometriosis and leiomyoma.

Author Response

Thank you very much for your comments. We feel very grateful of your review.

Reviewer 4 Report

A coll paper on a nice area : the Interstitial Cajal-like cells - here in the uterus.

History could be removed from the abstract.

The Figures , although very very nice, all 3 of them, they do have a single reference at the end - are they copied from there? modified? do you have the acceptance of those authors/?

I particularly enjoyed the conclusions section - very balanced and the perspectives - very very well done job here.

Author Response

Point 1: History could be removed from the abstract.

Response 1: We agree with you, and we have removed it from the abstract.

Point 2: The Figures, although very very nice, all 3 of them, they do have a single reference at the end - are they copied from there? modified? do you have the acceptance of those authors/?

Response 2: Thank you very much for this comment. All figures are original color drawings that are inspired on the histological data supported by the cited references. Figure 1 is based on a schematic black and white drawing made by Santiago Ramón y Cajal (1904), in which we have included the location of the ICCs at the two nervous plexuses. Figure 2 and 3 are schematic drawings inspired from two electron microscopy (figure 2) and inmunohistochemical (figure 3) images.

Thank you very much for your indications. Your review have greatly helped us to improve our manuscript.
